# Adherence to Hand Hygiene among Nurses and Clinicians at Chiradzulu District Hospital, Southern Malawi

**DOI:** 10.3390/ijerph191710981

**Published:** 2022-09-02

**Authors:** Monica Nzanga, Mindy Panulo, Tracy Morse, Kondwani Chidziwisano

**Affiliations:** 1Department of Environmental Health, Malawi University of Business and Applied Sciences, Private Bag 303, Chichiri, Blantyre 3, Malawi; 2Centre for Water, Sanitation, Health and Appropriate Technology Development (WASHTED), Malawi University of Business and Applied Sciences, Private Bag 303, Chichiri, Blantyre 3, Malawi; 3Department of Civil and Environmental Engineering, University of Strathclyde, Glasgow G1 1XQ, UK

**Keywords:** hand hygiene, adherence, nurse, clinician

## Abstract

Healthcare associated infections (HAIs) are a burden in many countries especially low-income countries due to poor hand hygiene practices in the healthcare settings. Proper hand hygiene in the healthcare setting is an effective way of preventing and reducing HAIs, and is an integral component of infection prevention and control. The objective of this study was to determine adherence to hand hygiene guidelines and associated factors among nurses and clinicians. A quantitative cross-sectional study was conducted at Chiradzulu District Hospital (Malawi) where stratified random sampling was used to obtain the sample of 75 nurses and clinicians. Data were collected using self-administered questionnaires (*n* = 75), observation checklists (*n* = 7) and structured observations (*n* = 566). The study findings confirmed low adherence to hand hygiene practice among healthcare workers (HCWs) in Malawi. Overall, higher hand hygiene practices were reported than observed among nurses and clinicians in all the World Health Organization’s (WHO) five critical moments of hand hygiene. This calls on the need for a combination of infrastructure, consumables (e.g., soap) and theory driven behavior change interventions to influence adoption of the recommended hand hygiene behaviors. However, such interventions should not include demographic factors (i.e., age, profession and ward) as they have been proven not to influence hand hygiene performance.

## 1. Introduction

Hospital acquired infections (HAIs) refer to the infections that patients develop either as the direct result of a healthcare intervention, or from contact with a healthcare setting [1]. Globally, HAIs affect approximately two million people annually with 5% to 10% requiring hospitalization [2], and these are twice high in low and middle income countries (LMICs) than High Income Countries [3]. Despite the scarcity of data on HAIs in LMICs, prevalence of HAIs in Sub-Saharan Africa have been reported to range from 1.6% to 28.7% with a prevalence of 13.4%, 11.4% and 8% in Botswana, Malawi and South Africa respectively [4,5,6]. HAIs pose a major threat to the safety of patients, guardians, and HCWs [7], with surgical site infections, urinary tract infections, pneumonia, and bloodstream infections being the common HAIs in LMICs [8]. It is estimated that approximately 50% of the HAIs are transmitted through the hands of HCWs while on duty [9]; which is of significant concern as this is preventable.

Studies have evidenced the importance of handwashing in disease reduction, with systematic reviews showing that handwashing with soap alone reduces diarrhea incidence by 30–47% [10,11]. Additionally, hand hygiene has been proven to control HAIs when conducted effectively at critical times [12]. Thus, the WHO promotes the “My five moments of hand hygiene” which encourage HCWs to clean their hands before patient contact, before an aseptic procedures, after exposure to body fluid, after patient contact, and after contact with patient’s surroundings [13]. However, adherence to hand hygiene among HCWs remains low with an average of 38.7% globally [13]. Malawi is no exception as the hand hygiene adherence rate was reported to be 23% at Queen Elizabeth Central Hospital (a major referral hospital) among clinicians and medical students [14]. This indicates that HAIs remain a public health burden in Malawian health care facilities. However, little is known about hand hygiene adherence and its associated factors among HCWs. Previous studies have primarily assessed hand hygiene adherence in healthcare setting using semi structured interviews or observations separately as data collection tools and they did not compare self-reported to actual behaviours [15,16,17]. However, the combination of both the semi structured interviews and observations have been successfully used to assess hand hygiene practice in community settings [18]. Thus, using both semi structured interviews and observations to understand hand hygiene practice in health care settings could strengthen the quality of data collected to inform design of interventions to improve hand hygiene behavior among the HCWs. This study aimed to assess both reported and observed hand hygiene practices among nurses and clinicians and associated factors in Southern Malawi.

## 2. Materials and Methods

### 2.1. Study Design and Setting

A cross-sectional study was conducted to assess hand hygiene adherence among nurses and clinicians from September to October 2019 (pre-COVID-19) at Chiradzulu District Hospital. The study was conducted at a district hospital considering that previous studies on hand hygiene practice related to HCWs were mainly conducted in tertiary hospital settings in urban areas within Malawi [14].

### 2.2. Study Population

We recruited nurses and clinicians on duty in all the wards at the hospital during the study period. Nurses and clinicians (who were off duty during data collection), medical students, laboratory technicians, and hospital attendants were excluded from the study.

### 2.3. Sampling and Sample Size

Stratified sampling was used to select a representative sample of the study participants from their working area (i.e., hospital wards). Random sampling was later used to obtain the proportional number of study participants from each ward. A sample size of 75 HCWs (38 nurses and 37 clinicians) from a total population of 126 HCWs (64 nurses and 62 clinicians) took part in the study. The sample size was obtained using a single population proportion formula [19] with a proportional *p*-value of 23% obtained from a study conducted at Queen Elizabeth Hospital [14], marginal error of 10%, 95% confidence interval, and a 10% non-response rate.

A recommendation by WHO of obtaining a minimum sample of 200 hand hygiene observations for each healthcare unit per observation period provided guidance on the number of observations conducted in this study [20].

### 2.4. Data Collection

Data were collected in three ways in the following order: (1) observations on the selected wards, (2) observation checklist and (3) self-administered questionnaire, 

Structured observations were used to determine the hand hygiene practice among nurses and clinicians indistinctly using the WHO five hand hygiene moments guideline [20]. Hand hygiene opportunities and hand hygiene events or actions with soap or alcohol-based hand rub (ABHR) were observed and recorded. “Hand hygiene opportunity refers to the moment during healthcare activities when hand hygiene was necessary to interrupt germ transmission through hands” [20]. The structured observation form was developed by modifying the handwashing audit form developed by the Sanitation and Hygiene Applied Research for Equity (SHARE) and the Centre of Water, Sanitation, Health and Appropriate Technology Development (WASHTED) [21]. In addition, data were collected using an observation checklist which was used to identify the availability of handwashing resources and cues to action including water, soap, ABHR, sinks, buckets, and posters in the hospital wards. A descriptive survey with a structured open and closed self-administered questionnaire was also used to capture social demographic factors, hand hygiene practices, and factors associated with adherence to hand hygiene guidelines among the study participants. The questionnaire was developed with reference to similar studies [22,23]. Hand hygiene observations among the HCWs were conducted two weeks before administering the questionnaires. Administration of questionnaires after the observations ensured that answered questions by participant did not influence the observations. The observations were conducted by one observer (i.e., one of the co-authors who is not a member of staff at Chiradzulu hospital) per time per ward from 8:00 a.m. to 12:00 p.m. and 1 p.m. to 4:00 p.m. for 14 days; 2 days in each of the observed seven wards at the hospital, namely, medical, surgical, pediatric, post-natal, Tuberculosis (TB), labor, and antenatal wards.

The collected data were coded, cleaned, and analyzed in the Microsoft Excel 2016 (Microsoft Corporation, Redmond, WA, USA) and STATA 14.1 (StataCorp.2015. Stata Statistical Software: Release 14. College Station, TX, USA: StataCorp. LP). Descriptive statistics were used to summarize the collected data and categorical variables were summarized using frequencies while means were used for continuous variables. The correlation test was used to find the relationship between the explanatory variables and the outcome. Chi-square test was used to find the difference between observed and reported hand hygiene practices while Wilcoxon-Mann-Whitney and Kruskal-Wallis tests were used to compare gender and profession of the study participant to hand hygiene adherence. Data collected through structured observations were used to find the hand hygiene adherence rate which was calculated by dividing the number of successful hand hygiene actions with soap or ABHR by the total number of observed hand hygiene opportunities. Hand hygiene adherence rate was also obtained from the reported data which was collected through the questionnaire. The hand hygiene practice score for each participant was calculated from the questions on the moments of hand hygiene which was later categorized into no (0–19%), low (20–49%), medium (50–79%), and high (80–100%) hand hygiene adherence.

### 2.5. Ethical Considerations

The study protocol was reviewed and approved by National Health Science Research Committee (NHSRC) (protocol #19/10/2408). A written authority to carry out the study was also obtained from Chiradzulu District Hospital. Additionally, questionnaire respondents provided informed consent for participation in the study.

## 3. Results

### 3.1. Social Demographic Factors

The mean age of the respondents was 27.88 ± 5.1 years. Most of the respondents worked in the post-natal ward (36%) likely because this ward had more nurses and clinicians compared to other departments that participated in the study (Table 1).

### 3.2. Observed Hand Hygiene Adherence

The hand hygiene observations revealed that nurses and clinicians had a combination of 566 hand hygiene opportunities (referring to the time one would be expected to perform hand hygiene practice) during the entire period of observations, with nurses having a higher number of high-risk interactions requiring hand hygiene (Table 2) Overall, low (22%) adherence to hand hygiene practice was observed, with higher levels among nurses than clinicians (Table 2). HCWs were seen to use none or sub-standard methods of hand hygiene (no hand hygiene action and handwashing with water only) at critical times (nurses: 73.3% and clinicians: 85.1%) (Table 2). When effective hand hygiene was performed, HCWs were observed using water and soap (14.7%) rather than ABHR (7.6%). This was attributed to the availability of water and soap compared to ABHR (Table 2). 

The study results show that high hand hygiene adherence was observed in the pediatric ward (41%) which also registered highest hand hygiene opportunities (114) more among nurses (35%) than clinicians (6%). The least hand hygiene performance was observed in the antenatal ward (Table 3).

### 3.3. Comparison of the Observed and Reported Hand Hygiene Practice

The findings indicated that there was a significant difference (*p*-value = 0.0001) between the low observed hand hygiene adherence and the self-reported adherence by the study participants across all the critical moments of hand hygiene (Table 4). 

Furthermore, observed and reported hand hygiene practices were compared between nurses and clinicians to complement the overall difference. A significant difference was noted between the reported and observed hand hygiene practices among nurses (*p*-value = 0.0001) and clinicians (*p*-value < 0.0028) in all the critical moments of hand hygiene except before an aseptic procedure among clinicians (*p*-value = 0.1061) (Table A3 in Appendix A).

### 3.4. Hand Hygiene Knowledge

Using selected key hand hygiene attributes, hand hygiene knowledge was assessed among study participants (Table 5). The majority of the study participants (92%) knew about hand hygiene guidelines primarily from university/college studies as hygiene/Infection Prevention and Control were fundamental components of their training programs. However, few received formal refresher trainings on hand hygiene, with the majority receiving the training at least 2 years ago (Table 5). Few participants were able to describe the appropriate technique for either handwashing with soap (24%) or ABHR (38.5%) (Table 5).

### 3.5. Relationship between Demographic Factors and Hand Hygiene Adherence

The study found that there was no significant relationship between age and adherence to hand hygiene (*p*-value = 0.09). The Wilcoxon-Mann-Whitney test showed no significant difference between reported hand hygiene adherence and gender of the HWCs (*p*-value = 0.29) (Table A1). Using the Wilcoxon-Mann-Whitney test, a comparison between the cadres (i.e., nurses and clinicians) did not find a significant difference between hand hygiene adherence and profession (*p*-value = 0.23) (Table A1). Similarly, Kruskal-Wallis equality-of-populations rank test showed no difference between hand hygiene adherence and the ward unit in which the study participants worked (*p*-value = 0.39) (Table A2).

### 3.6. Barrier to Hand Hygiene Adherence among the Healthcare Workers

Inadequate hand hygiene resources were reported as the main reason for not performing hand hygiene by most of the study participants (54.7%). This was evidenced by the absence of functioning sinks with a tap, soap, ABHR and posters in wards during the 5 day observations as shown in Table 6. However, despite having almost similar resources for hand hygiene in some wards (i.e., pediatric, post-natal and labor wards), hand hygiene remains high in the pediatric ward compared to the other wards (Table 6).

In addition, participants reported high workload (42.7%), negligence (4%) and forgetfulness (5.3%) as other reasons that prevented them from performing hand hygiene. The study also found that the use of gloves acted as a barrier to hand hygiene as most nurses (26.4%) and clinicians (14.5%) were observed not performing hand hygiene after using gloves. Nevertheless, study participants reported that infection prevention (48%), prevention of cross-contamination (25.3%), and availability of resources (10.7%) motivated them to perform hand hygiene. However, the hospital had no specific interventions to support the practice of hand hygiene among the HCWs. In addition, the nurses and clinicians were of the view that the improvement in the availability of hand hygiene resources (62.7%), provision of training (50.7%), recruitment of more staff (24%), and motivation in form of incentives (i.e., awards) for a best hand hygiene performer (12.4%) can increase their adherence to hand hygiene.

## 4. Discussion

Adherence to the WHO’s “my five critical moments of hand hygiene” guidelines is vital in the prevention and control of HAI among the service providers, patients, and their guardians in the health care setting [13]. The current study measured hand hygiene adherence and its associated factors among the health care workers at Chiradzulu District Hospital in Malawi; a different setting from the other studies conducted in the country where the assessment was from the tertiary health care facilities.

The study established low level of hand hygiene adherence among the HCWs in Chiradzulu District Hospital; coincidentally, this was similar to what was noted at one of the major referral hospitals in Malawi [14] and other LMICs [24,25]. This implies that the hand hygiene behavior among HCWs is almost the same despite the setting of their working place i.e., tertiary hospital or district hospital.

Related to other studies conducted in healthcare settings [26,27], our study findings indicated high self-reported hand hygiene adherence than the observed ABHR and handwashing with soap and water behavior. This shows that HCWs are aware about the need to practice hand hygiene; however, such knowledge is not fully translated into practice. Smith et al. (2019) confirmed the need for theory driven behavior change interventions to improve hand hygiene behavior in healthcare settings [28], and a systematic review done by Srigley et al. (2015) showed sustained improved hand hygiene practice among HCWs/patients/guardians when they were exposed to psychological frameworks/theories of behavior change interventions [29]. Absence of specific hygiene promotion interventions at Chiradzulu district hospital during the study period confirmed a lack of the theory driven behavior change interventions. 

Failure of the study participants to practice hand hygiene due to limited or unavailability of water, sanitation and hygiene (WASH) infrastructure and materials support what others have reported that presence of these facilities (e.g., handwashing facility with soap) is a predictor of handwashing behavior [30]. As indicated elsewhere [31,32,33], WASH infrastructure promotes hand hygiene adherence as observed in our study where pediatric ward had a higher hand hygiene adherence rate as it had most of the required hand hygiene resources compared to other wards. Thus, environmental modifications such as provision of adequate handwashing facilities with soap and water including presence of ABHR have the potential to improve the hand hygiene practice. In addition, use of environmental prompts and cues for action such as posters have proven to be effective motivators to hand hygiene practice among HCWs [34]. 

Studies conducted in Japan, Ethiopia and United States of America preferred using ABHR to handwashing with water and soap due to its shorter time to disinfect the hands [24,25,35,36]. This shows the similarity in preference of use of ABHR in both LMICs and high income countries implying that economic difference of countries does not affect the preference of ABHR to handwashing among HCWs. However, our study findings established that most of the study participants practiced hand hygiene using water and soap due to unavailability of the ABHR in the wards. Considering HCW to patient ratio of 1.44 per 1000 population in Malawian healthcare facilities [37] which is contrary to WHO recommendation of 4.45 per 1000 population [38], availability of ABHR is vital to improve hand hygiene behavior among HCWs since it is easy to use between patients. Nevertheless, there is a need to find out the preference between use of ABHR and handwashing with soap and water among Malawian HCWs. 

WHO recommends five critical moments of hand hygiene in the healthcare settings which include practice of hand hygiene after touching a patient [20]. However, our study observed an increase in hand hygiene practice before rather than after touching a patient; a situation which may facilitate cross-contamination [9]. A study in Ghana observed that HCWs performed hand hygiene after touching a patient as the risk of cross-contamination was perceived to be high [16]. Inadequate hand hygiene practice by HCWs after touching a patient could be due to their perception that they were less vulnerable to infection transmission since they used gloves during service delivery. Thus, prominence should be given to the importance of hand hygiene despite the use of gloves.

As reported in other studies [27,39], we did not find a relationship between hand hygiene adherence and demographic factors (age, profession, and ward). Contrary to a study which was conducted at Kenyatta National Hospital which found that there was a statistically significant relationship between most of the demographic factors and compliance to infection prevention and injection safety practices among nurses [40]. Our study finding is different from this and other studies, hence there is a need for further exploration to understand why there is no significant relationship between demographic factors and adherence to hand hygiene at Chiradzulu district hospital. The absence of a relationship between demographic factors and adherence to hand hygiene in this study, implies that the selected demographic factors have no influence on hand hygiene practices among HCWs. Thus, designing of behavior change interventions to improve hand hygiene practices among HCWs should not focus on these demographic factors. 

Despite HCWs reporting high compliance for their hand hygiene practice, a knowledge assessment using selected key hand hygiene attributes indicated that majority of the HCWs had low knowledge about hand hygiene. The study has established that Chiradzulu district hospital had no specific program to raise hand hygiene awareness among the HCWs; thus, it is assumed that the HCWs gained hand hygiene knowledge during their formal training in college or university which they are prone to forget with the passage of time. Hence, there is need to provide regular refresher hand hygiene trainings to HCWs as it has been proven to improve hand hygiene knowledge among HCWs [9,41]. Importantly, integrating hand hygiene awareness activities into existing hospital-based programs (e.g., clinical dialogue sessions and continuous professional development sessions) would ensure sustained discussion about the topic; and thus, serve as a reminder to the HCWs on the importance of adhering to recommended hand hygiene guidelines in all the five critical moments of hand hygiene.

### Limitations

Use of observations subjected the study participants to Hawthorne effect [42]. However, the Hawthorne effect was reduced by the use of questionnaires to triangulate the collected data. One observer was used in observing the hand hygiene practices which could have resulted into missing out other important opportunities. Nevertheless, this was reduced by conducting observations for 14 days. The study was conducted at one district hospital therefore the findings cannot be generalized to all district hospitals in Malawi, hence there is a need to confirm the findings of this study in other district hospitals. 

## 5. Conclusions

This study confirmed low adherence to hand hygiene practice among HCWs in Malawi. Further, our findings concur with other studies that presence of WASH infrastructure promotes performance of recommended hand hygiene behavior. Importantly, high self-reported than observed hand hygiene practices by the HCWs confirmed the need for theory driven behavior change interventions to influence adoption of the recommended hand hygiene behavior. Such interventions should include prominence on hand hygiene in all the five critical moments of hand hygiene despite the use of gloves by HCWs. In addition, there should be regular engagement with the HCWs (e.g., through refresher trainings and use of already existing dialogue sessions) to strengthen the hand hygiene behavior. When designing such interventions, demographic factors (i.e., age, profession, and ward) should not be considered as they have proven not to influence hand hygiene behavior performance.

## Figures and Tables

**Table 1 ijerph-19-10981-t001:** Demographic factors of the study participants.

Variables		Gender		Total *n* (%)
		Male *n* (%)	Female *n* (%)	
Age	20–29 years	24 (32)	29 (39)	53 (71)
	30–39 years	14 (19)	7 (9)	21 (28)
	40 years and above	0 (0)	1 (1)	1 (1)
Profession	Nurse	13 (17)	25 (34)	38 (51)
	Clinician	23 (30)	14 (19)	37 (49)
Ward	Medical	10 (13)	5 (7)	15 (20)
	Surgical	9 (12)	8(11)	17 (23)
	Pediatic	6 (8)	7 (9)	13 (17)
	Post-natal	10 (13)	17 (23)	27 (36)
	TB	1 (1)	0 (0)	3 (4)
	Labour	0 (0)	0 (0)	0 (0)
	Antenatal	1 (1)	1 (1)	2 (3)

Tuberculosis (TB).

**Table 2 ijerph-19-10981-t002:** Hand hygiene adherence to soap and hand rub among nurses and clinicians.

Professional	Opportunities	No Adherence to Hand Hygiene, *n* (%)	Adherence to Handwashing with Water Only, *n* (%)	Adherence to Handwashing with Soap and Water, *n* (%)	Adherence to ABHR *n* (%)	Total Adherence, *n* (%)
Overall	566	391(69.1)	48(8.5)	83 (14.7)	43 (7.6)	126 (22)
Nurse	352	220 (62.5)	38 (10.8)	62 (17.6)	32 (9.1)	94 (27)
Clinician	214	172 (80.4)	10 (4.7)	21 (9.8)	11 (5.1)	32 (15)

Alcohol-based hand rub (ABHR).

**Table 3 ijerph-19-10981-t003:** Hand hygiene adherence in the hospital wards.

Ward	Opportunity (*n*)		Hand Hygiene Action Taken *n* (%)		Total Adherence (%)
	Nurse	Clinician	Nurse	Clinician	
Medical	101	73	17(10)	12(7)	29(17)
Surgical	67	43	11(10)	7(6)	18(16)
Pediatric	114	36	53 (35)	9 (6)	62 (41)
Post-natal	24	18	10 (24)	1 (2)	11 (26)
TB	2	4	0 (0)	1 (17)	1 (17)
Labour	30	6	5 (14)	3 (8)	8 (22)
Antenatal	14	34	0 (0)	0 (0)	0 (0)

Tuberculosis (TB).

**Table 4 ijerph-19-10981-t004:** Observed versus reported hand hygiene practice at different critical moments of hand hygiene.

Hand Hygiene Critical Moments	Reported Hand Hygiene Practice *n* (%)	Observed Hand Hygiene Practice *n* (%)	X^2^ Test (*p*-Value)
Before touching a patient	52 (69)	30 (13)	0.0001
Before aseptic procedure	66 (88)	9 (19)	0.0001
After body fluid	71 (95)	35 (41)	0.0001
After touching a patient	60 (80)	52 (33)	0.0001
After touching a patient’s surrounding	51 (68)	0 (0)	0.0001

**Table 5 ijerph-19-10981-t005:** Variables on hand hygiene knowledge.

Variable	Response	Percentage (%)
Hand hygiene guidelines	Yes	92
Formal training	Yes	16
Trainer	MoH	100
When was the training	3 months ago	18.2
	6 months ago	18.2
	1 year ago	27.3
	2 years and above	36.4
Routine Use of ABHR	Yes	28
Required time for ABHR (within 20–30 s)	Yes	38.5
Handwashing technique	Yes	24

Ministry of Health (MoH), ABHR (alcohol-based hand rub).

**Table 6 ijerph-19-10981-t006:** Availability of hand hygiene resources.

Ward	Poster	Sink		Bucket		ABHR	
		Soap	Water	Soap	Water	Ward	Personal
Medical	0	*5*	*2*	*5*	*5*	0	0
Surgical	0	0	0	*5*	*5*	0	0
Pediatric	0	*5*	*2*	*5*	*5*	0	0
Post-natal	0	*5*	*3*	*5*	*5*	0	0
TB	0	0	0	0	*5*	0	0
Labour	1	0	*3*	*5*	*5*	0	0
Antenatal	0	0	*4*	0	0	0	0

Tuberculosis (TB), Alcohol-based hand rub (ABHR).

## Data Availability

All data referred to within this study is available upon request.

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
