# Peer review of "Adherence to Hand Hygiene among Nurses and Clinicians at Chiradzulu District Hospital, Southern Malawi"

_ijerph, 2022, doi:10.3390/ijerph191710981_

Round 1
Author Response
Thank you very much for the constructive comments. Attached documents contains how your comments have been addressed.
Regards,
Monica.

Reviewer 2 Report
Dear authors,
after reviewing your manuscript, I consider that although the main results are already known through other published works, your manuscript is quite interesting. Your contributions allow to increase the knowledge about adherence to hand hygiene in countries or regions less analyzed, increasing the world knowledge on this topic.
However, major changes have to be made so that the manuscript increases its scientific quality to be published. I encourage authors to implement the following changes:
WRITING STYLE REVIEW
* There are important inconsistencies in the way you introduce abbreviations. For instance, in abstract you introduce the abbreviation HCWs but there is no previous explanation, as in other introduced abbreviations. On the contrary in line 34 you introduce "HICs" (correctly explained) but no more HICs appear in the text, so maybe it is not necessary to include the abbreviation.
In addition to this, please review the following abbreviations in the whole manuscript:
*ABHR: it appears irregularly as "alcohol-based hand rub" or as ABHR, please, unify the criteria (please consult: https://www.editage.com/insights/common-errors-usage-abbreviations-scientific-writing)
* HH (for hand hygiene): it appears one unique time as abbreviation in the whole manuscript (consider to generalize it).
There are other writing problems that need to be solved:
*P-value (in capital letter): correct form: p-value or p value
*Please unify criteria in the writing of cross-contamination (lines 204, 258) vs. cross contamination (line 259). Also, hand washing (with space) (line 236) vs. handwashing (no space) (line 241)
*In the discussion section (lines 235 and 236) you introduce the term WASH but it is not clear if it refers to WASTHED (line 102). If it refers to WASTHED, why you change the abbreviation?
*Why you write Hand (with capital letter) hygiene in line 259?
*There are also problems in the reference of in-text citations. Please, review how to write the name of a specific author (et al). In-text citation of J Smith et al (line 227) and Srighley Jet et (line 229) is not correct at all.
TABLES AND FIGURES REVIEW
*Please review the number of tables and figures included in the manuscript. After table 4 it follows table 8 (there is no tables 5, 6 and 7).
*All figures and tables need to include as foot of table or figure what included abbreviations mean. E.g., table 4 include MoH but it is not explained what it means, also TB and ABHR in "table 8". Each table should be self-explanatory on its own, without the need to read the text.
*Although all figures and tables are referred in text, figure 1 is not referred in any paragraph. Please correct the mistake.
*Figure 2 does not provide any relevant information. I recommend the authors remove it.
*Table 1 contains interesting information but no interline space are between variables included in profession and ward. It is a bit confusing to understand where ends and where begin the different items.
*Table 3. If you write always nurse and clinicians (in this order) in the whole text and tables/figures it is better to maintain the same way to write it to unify criteria.
CONTENT REVIEW
After reading the manuscript, there is information that it is not well explained or can be reviewed to avoid inconsistencies.
*Line 43: you indicate that studies evidencing the importance of handwashing are recent, but the effectiveness of this technique to avoid infections is dated from XIX century when Ignaz Semmelweis discovered the effectiveness of hand washing. I encourage authors to change "recent studies" to another more realistic sentence.
*Line 69: although information of Chiradzulu district is interesting, I hesitate if the information about the population number is relevant in this paragraph (the study results will not change independently of the citizens population in the district).
*Lines 106-112: perhaps you can highlight if the observer recorded the moments of handwashing in the nurses, in the clinicians indistinctly at the same moment or if, on the contrary, he made the observations following some characteristics in the nurses or in the clinicians.
*Lines 158-165 (3.3 section): this is probably the paragraph that need to be reviewed more deeply. You indicate that clinicians perform more hand hygiene than nurses in general, but looking at figure 1 that is not really true. Nurses perform more hand hygiene before and after touching a patient (an action most often performed by nurser), while clinicians do the same before an antiseptic procedure (an action that in some contexts perform more the clinicians). I encourage the authors to clarify the differences between nurses and clinicians.
*Line 178: I am wondering if 69% can be considered "the majority of the study participants"...
In the discussion section you compare your results with those obtained in Japan, Ethiopia and USA. I think it should be discussed if those differences are due to the similarities/differences between Low-Middle Income Countries (Ethiopia) and High-Income Countries (Japan and USA) or economic incomes of a country does not imply any difference.
Finally, although in your study did not observe any statistical significance in age or gender and the performance of hand hygiene, several studies indicate the contrary: there are studies (including health professionals or even health professionals’ students) that observe significative differences in age and gender. I encourage the authors to discuss why in your study do not observe differences but there are those differences in other works.
I want to thank the authors to improve their manuscript reviewing and including the proposed changes. I am sure that your work will increase the scientific quality.
Author Response
Thank you very much for the constructive comments, they have really sharpen our manuscript. Attached documents contains how your comments have been addressed.
Regards,
Monica.
